# Estimating economic and disease burden of snakebite in ASEAN countries using a decision analytic model

Chanthawat Patikorn[1], Jörg Blessmann[2], Myat Thet Nwe[3], Patrick Joseph G. Tiglao[4,5,6,7], Taksa Vasaruchapong[8], Tri Maharani[9], Uyen Vy Doan[10], Syafiq Asnawi Zainal Abidin[11], Ahmad Khaldun Ismail[12], Iekhsan Othman[11], Suthira Taychakhoonavudh[1]*, Nathorn Chaiyakunapruk[13,14,15]*

1 Department of Social and Administrative Pharmacy, Faculty of Pharmaceutical Sciences, Chulalongkorn University, Bangkok, Thailand, 2 Department of Implementation Research, Bernhard Nocht Institute for Tropical Medicine, Hamburg, Germany, 3 Myanmar Snakebite Project, Mandalay, Myanmar, 4 Department of Emergency Medicine, Eastern Visayas Regional Medical Center, Tacloban City, Philippines, 5 Philippine Toxinology Society, Incorporated, Manila, Philippines, 6 Department of Emergency Medicine, University of the Philippines-Manila, Philippine General Hospital, Manila, Philippines, 7 Department of Emergency Medicine, Corazon Locsin Montelibano Memorial Regional Hospital, Bacolod City, Negros Occidental, Philippines, 8 Snake Farm, Queen Saovabha Memorial Institute, The Thai Red Cross Society, Bangkok, Thailand, 9 National Institute Research and Development, Ministry of Health, Jakarta, Indonesia, 10 Division of Medical Toxicology, Cho Ray Hospital, Ho Chi Minh City, Vietnam, 11 Jeffrey Cheah School of Medicine and Health Sciences, Monash University Malaysia, Bandar Sunway, Selangor, Malaysia, 12 Department of Emergency Medicine, Faculty of Medicine, Universiti Kebangsaan Malaysia, Bandar Tun Razak, Kuala Lumpur, Malaysia, 13 Department of Pharmacotherapy, College of Pharmacy, University of Utah, Salt Lake City, Utah, United States of America, 14 IDEAS Center, Veterans Affairs Salt Lake City Healthcare System, Salt Lake City, Utah, United States of America, 15 School of Pharmacy, Monash University Malaysia, Selangor, Malaysia

* suthira.t@pharm.chula.ac.th (ST); nathorn.chaiyakunapruk@utah.edu (NC)

**Data Availability Statement:** Data supporting the findings of this study are available within the article and its supporting information.

## Abstract

### Background

Understanding the burden of snakebite is crucial for developing evidence-informed strategies to pursue the goal set by the World Health Organization to halve morbidity and mortality of snakebite by 2030. However, there was no such information in the Association of Southeast Asian Nations (ASEAN) countries.

### Methodology

A decision analytic model was developed to estimate annual burden of snakebite in seven countries, including Malaysia, Thailand, Indonesia, Philippines, Vietnam, Lao PDR, and Myanmar. Country-specific input parameters were sought from published literature, country's Ministry of Health, local data, and expert opinion. Economic burden was estimated from the societal perspective. Costs were expressed in 2019 US Dollars (USD). Disease burden was estimated as disability-adjusted life years (DALYs). Probabilistic sensitivity analysis was performed to estimate a 95% credible interval (CrI).

**Funding:** This work is supported by the Wellcome Trust [218539/Z/19/Z] to CP, AKI, IO, SAZA, ST, and NC (https://wellcome.org). This research project is supported by the Second Century Fund (C2F), Chulalongkorn University to CP and ST (https://c2f.chula.ac.th). The funders had no role in study design, data collection, data analysis, data interpretation, writing of the report, or the decision to submit for publication.

**Competing interests:** The authors have declared that no competing interests exist.

## Principal findings

We estimated that annually there were 242,648 snakebite victims (95%CrI 209,810–291,023) of which 15,909 (95%CrI 7,592–33,949) were dead and 954 (95%CrI 383–1,797) were amputated. We estimated that 161,835 snakebite victims (69% of victims who were indicated for antivenom treatment) were not treated with antivenom. Annual disease burden of snakebite was estimated at 391,979 DALYs (95%CrI 187,261–836,559 DALYs) with total costs of 2.5 billion USD (95%CrI 1.2–5.4 billion USD) that were equivalent to 0.09% (95% CrI 0.04–0.20%) of the region's gross domestic product. >95% of the estimated burdens were attributed to premature deaths.

## Conclusion/Significance

The estimated high burden of snakebite in ASEAN was demonstrated despite the availability of domestically produced antivenoms. Most burdens were attributed to premature deaths from snakebite envenoming which suggested that the remarkably high burden of snakebite could be averted. We emphasized the importance of funding research to perform a comprehensive data collection on epidemiological and economic burden of snakebite to eventually reveal the true burden of snakebite in ASEAN and inform development of strategies to tackle the problem of snakebite.

## Author summary

In the Association of Southeast Asian Nations (ASEAN) countries, we estimated that annually there were 242,648 snakebite victims of which 15,909 victims were dead and 954 victims were amputated. We estimated that 69% of victims indicated for antivenom treatment were not treated with antivenom. Annual disease burden of snakebite was estimated at 391,979 disability-adjusted life years (DALYs) with estimated total costs of snakebite of 2.5 billion US Dollars which were equivalent to 0.09% of the region's gross domestic product. Almost all of the estimated economic and disease burdens were related to deaths from snakebite envenoming which suggested that the remarkably high burden of snakebite could actually be averted. We emphasized the importance of funding research to perform a comprehensive data collection on epidemiological and economic burden of snakebite to eventually reveal the true burden of snakebite in ASEAN and inform development of strategies to tackle the problem of snakebite.

## Introduction

Snakebite is a neglected tropical disease that was estimated to affect 5.4 million victims with up to 138,000 deaths around the world [1]. Snakebite envenoming has been recognized by the World Health Organization (WHO) as the highest priority neglected tropical diseases since 2017. The WHO has set its goal to halve the global morbidity and mortality burden of snakebite by 2030 [2, 3].

The Association of Southeast Asian Nations (ASEAN) is an economic union comprising of ten member countries including Brunei Darussalam, Cambodia, Indonesia, Lao PDR, Malaysia, Myanmar, Philippines, Singapore, Thailand, and Vietnam with over 600 million people

[4]. ASEAN is one of the tropical regions with disproportionately high incidence of snakebite. Previous estimation of snakebite in 2007 found that approximately 234,000–1,410,000 people were bitten by snake annually resulting in 700–18,000 deaths in eight ASEAN countries, except Brunei Darussalam and Singapore where snakebite rarely occurred and/or exact data were lacking [1].

Our previous study found that there are five domestic antivenom manufacturers in ASEAN, including Thailand, Indonesia, Philippines, Vietnam, and Myanmar. Up to 290,000 vials of antivenoms were annually produced by these manufacturers which could treat approximately 42,000 victims with snakebite envenoming. However, these produced antivenoms were not enough to treat all victims indicated for antivenom treatment. Besides, the total demand of antivenoms in ASEAN was not estimated [5]. This warranted a comprehensive research on burden of snakebite in the region to quantitatively highlight the neglected problem.

Understanding the current economic and disease burden of snakebite is crucial for developing evidence-informed strategies to reduce morbidity and mortality of snakebite victims to pursue the goal set by the WHO [3]. Studies have been conducted to estimate the annual national economic and disease burden of snakebite in regions where snakebites are prevalent such as Africa [6–12]. Nevertheless, there was no such information in ASEAN countries. Thus, we aimed to estimate economic and disease burden of snakebite in ASEAN using a decision analytic modelling approach.

## Methods

### Ethics statement

This study was approved by the Monash University Research Ethics Committee (Project ID: 23246). Written consent was formally obtained from the participants.

### Overall approaches

A decision analytic model was developed to estimate the annual economic and disease burden of snakebite in seven ASEAN countries including Malaysia, Thailand, Indonesia, Philippines, Vietnam, Lao PDR, and Myanmar. These seven countries were selected based on the evidence of documented snakebite in the country and availability of local key informants to gather more insights on the situation of snakebite which were not publicly available. Brunei Darussalam and Singapore were not included because snakebite rarely occurred and/or exact data were lacking [1]. Cambodia was not included due to lack of recently published literature on snakebite and key informants that hindered the proper estimation of the burden of snakebite in Cambodia.

Annual number of snakebite victims in the region were estimated using a decision analytic model which incorporated treatment seeking behavior to include victims who were not treated in healthcare facilities. Economic burden was estimated from the societal perspective to estimate lifetime costs of snakebite victims which occurred from snakebite episode to long-term consequences. To enable comparison of economic burden between countries, all costs of snakebite were presented as annual national total costs for each country in 2019 USD and converted to the percentage of country's gross domestic product (GDP) in 2019. Disease burden of snakebite was estimated and quantified as disability-adjusted life years (DALYs) lost due to snakebite in one year in each country.

### Decision analytic model

A decision analytic model was developed to simulate the course of snakebite victims in ASEAN which was adapted from previous economic evaluations of antivenoms for snakebite

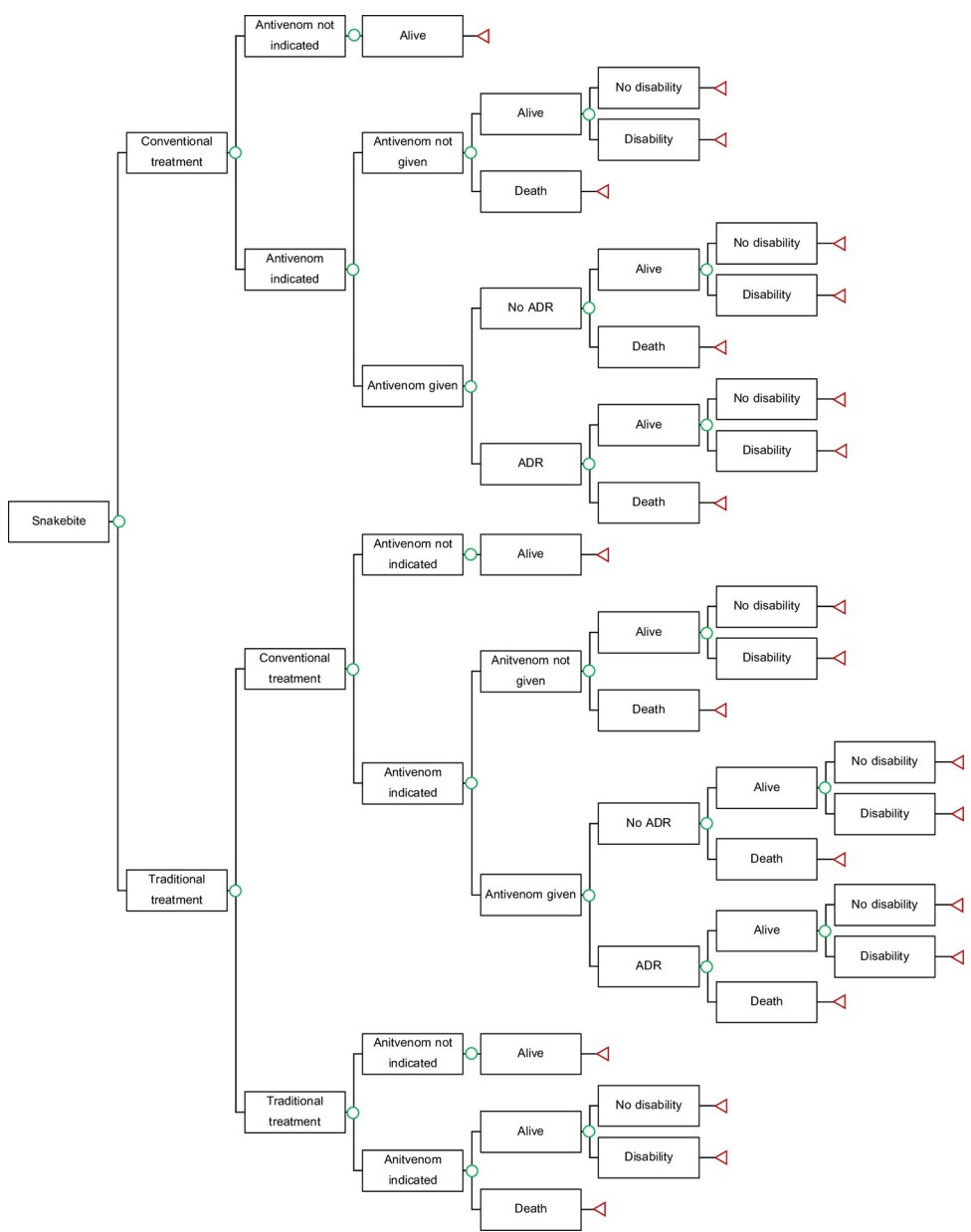

**Fig 1. Decision tree to estimate economic and disease burden of snakebite in ASEAN countries.** Abbreviation: ADR–adverse drug reaction.

antivenom in West Africa (**Fig 1**) [13, 14]. Victims who were bitten by snake sought for treatment either at conventional treatment (hospitals or healthcare facilities) or traditional treatment through traditional healers to reflect the treatment seeking behavior of victims in the region [5]. Victims who firstly sought traditional healers might subsequently switch to conventional treatment or continue their treatments with traditional healers. Snakebite victims might be indicated for antivenom treatment depending on the occurrence of systemic envenoming following snakebite. Victims who were not indicated for antivenom treatment were assumed to result in being alive as the envenoming is not life-threatening [15–21]. Victims indicated for antivenom treatment who sought conventional treatment might be given with antivenom

depending on the current level of access to antivenom in each country. Level of access to antivenom was determined by the number of antivenoms treatment available divided by number of victims indicated for antivenom treatment. Victims who received antivenom treatment might have adverse drug reaction (ADR) following antivenom administration. Victims indicated for antivenom treatment regardless of their treatment seeking behavior might be alive or dead. Alive victims might have disability. Disabilities included in this model were digit and limb amputation.

## Input parameters

Country-specific input parameters were sought from various sources, including published literature, data from the country's Ministry of Health, local data, and expert opinion (**S1 Table**) [15–62]. An in-depth interview with key informants who were experts in snakebite in ASEAN was also conducted to confirm the retrieved parameters, refer to potential sources of information that might not be publicly available, and ask for their opinions when data were not available. The input parameters were validated through triangulation of data from literature, local data, and interview. Justification of input parameters was described in **S1 Appendix**.

Main sources of information were national statistics and published research for the burden estimation of Malaysia, Thailand, and Myanmar. Published research and anecdotal evidence (local data, and expert opinion) were the main sources of information for the burden estimation of Vietnam, and Lao PDR. Anecdotal evidence was the only source of information for the burden estimation of Indonesia, and Philippines.

## Model assumptions

There were three key assumptions of the model. First, one person can be bitten by snake only once in a lifetime. Second, snakebite victims were accompanied by relatives or family members who took care of them during snakebite episode. Third, antivenom was given to reverse snakebite envenoming and save lives. However, there was no data on the efficacy or effectiveness of antivenom in ASEAN countries. Thus, antivenom effectiveness was based on a study in Nigeria which found a 2.33 fold (95% confidence interval [CI]; 1.26–4.06) increase risk of death in antivenom indicated victims who were not treated with antivenom compared to those treated with antivenom [40]. This relative risk was used to calculate the probability of death due to snakebite in those who were not treated with geographically appropriate antivenoms.

## Total number of snakebite victims

Estimating the total number of snakebite victims occurring in one year in each country was done by applying the country-specific input parameters into the model. The estimated snakebite victims were categorized by their gender, age groups, treatment seeking behavior, indication for antivenom treatment, and disease consequences, i.e., deaths, alive without disabilities, and alive with disabilities.

## Costs of snakebite

Costs of snakebite were estimated from societal perspective, including direct medical costs, direct non-medical costs, and indirect costs (**S1 Table** and **S1 Appendix**). Direct medical costs were estimated using a bottom-up approach which included costs of hospitalization, antivenom treatment, antivenom logistics, ADR management, and amputation. Direct non-medical costs included costs of transportation and additional food for victims and their relatives during snakebite episodes. Indirect costs were estimated using a human capital approach by

multiplying the time lost due to illness to the daily income based on the GDP per capita of each country [61]. Indirect costs included productivity losses during snakebite episode of victims and their relatives and productivity losses due to premature death. Productivity losses during snakebite episodes for victims and their relatives were estimated by multiplying length of stay to the daily income. Productivity losses due to premature death were estimated by multiplying the remaining working years from the age of death up to retirement age at 60 years to the GDP per capita. Productivity losses were not quantified for those who died after the age of 60. Productivity losses due to premature death were discounted at the rate of 3% and adjusted for annual growth of GDP per capita in each country [58–60, 62].

## Disease burden of snakebite

Disease burden of snakebite was calculated as DALYs using the template developed by WHO [63]. DALYs were the sum of years of life lost (YLL) and years lived with disability (YLD). YLLs due to snakebite envenoming were calculated from the number of deaths multiplied by a global standard life expectancy at the age of death. YLDs of snakebite victims included YLDs for snakebite episode and YLDs for amputation. YLDs were calculated from the duration of disability multiplied to a disability weight for each condition according to the Global Burden of Disease 2013 study (**S1 Table**) [46].

## Analysis

Economic and disease burden of snakebite in ASEAN was estimated using input parameters as base-case estimates. Sensitivity analyses were performed to assess the model robustness. One-way sensitivity analysis was performed to assess uncertainty of the base-case input parameters over their plausible ranges on the model outputs. Scenario analysis was performed by incorporating post-traumatic stress disorder (PTSD) into the model as a mental disability which estimated that PTSD would occur in 8% (95%CI; 2–18%) of the victims who survived from snakebite envenoming [64]. PTSD could also occur following a snakebite without systemic envenoming. However, the incidence was unknown. Therefore, by applying a lower boundary level of the probability of PTSD following snakebite, it was estimated that 2% of snakebite victims without envenoming would have PTSD following snakebite. Estimation of economic burden of PTSD following snakebite is explained in **S2 Appendix** [65–67]. Probabilistic sensitivity analysis was performed using Monte Carlo simulations for 1,000 times by randomly sampling on a distribution of all parameters to estimate a 95% credible interval (CrI) of the economic and disease burden of snakebite.

## Patient and public involvement

Patients or the public were not involved in the design, or conduct, or reporting, or dissemination plans of our research.

# Results

## Snakebite victims in ASEAN

The model estimated that there were 242,648 snakebite victims (95%CrI 209,810–291,023) annually occurring in ASEAN with annual incidence of 38.03 per 100,000 population (95%CrI 32.89–45.62). The estimated incidence of snakebite ranged from the lowest in Malaysia (10.68 per 100,000 population) to the highest in Lao PDR (200.00 per 100,000 population). (**Tables 1** and **S2**).

**Table 1. Estimated annual disease burden of snakebite in ASEAN countries.**

| | Snakebite victims, n | Antivenom indicated victims, n | Deaths, n | Amputations, n | YLLs | YLDs | DALYs | DALYs per 100,000 population |
|---|---|---|---|---|---|---|---|---|
| **Malaysia**[*] | 3,412 (3,303–3,533) | 481 (254–767) | 2 (0–6) | 0 | 50 (0–151) | 1.4 (0.6–2.5) | 52 (1–152) | 0.2 (0.003–0.5) |
| **Thailand**[*] | 8,715 (8,525–8,906) | 5,166 (3,766–6,482) | 4 (2–7) | 2 (0–7) | 102 (51–178) | 8 (4–14) | 110 (57–185) | 0.2 (0.1–0.3) |
| **Indonesia**[+] | 135,000 (134,297–135,689) | 49,632 (34,229–65,496) | 10,547 (5,012–22,563) | 799 (355–1,426) | 262,302 (124,650–561,145) | 586 (246–1,120) | 262,888 (125,252–562,144) | 97 (46–208) |
| **Philippines**[+] | 13,377 (11,452–15,772) | 1,755 (1,457–2,127) | 550 (274–1,099) | 12 (6–16) | 13,311 (6,624–26,641) | 7 (4–11) | 13,320 (6,632–26,649) | 12 (6–25) |
| **Vietnam**[¶] | 46,745 (17,500–91,013) | 41,236 (15,290–80,701) | 1,655 (490–4,440) | 0 | 40,136 (11,869–107,679) | 114 (38–258) | 40,250 (11,931–107,876) | 42 (12–112) |
| **Lao PDR**[¶] | 14,339 (14,111–14,571) | 3,029 (2,917–3,138) | 1,007 (510–2,009) | 141 (22–348) | 24,468 (12,420–48,837) | 61 (10–189) | 24,532 (12,462–48,880) | 342 (174–682) |
| **Myanmar**[*] | 21,059 (20,623–21,540) | 16,275 (15,877–16,679) | 2,145 (1,303–3,824) | 0 | 50,786 (30,877–90,632) | 44 (27–67) | 50,830 (30,926–90,673) | 94 (57–168) |
| **Total** | 242,648 (209,810–291,023) | 117,575 (73,790–175,390) | 15,909 (7,592–33,949) | 954 (383–1,797) | 391,154 (186,491–835,263) | 825 (329–1,661) | 391,979 (187,261–836,559) | 61 (29–131) |

Estimates are presented as base-case estimates with their 95% credibility interval (in parentheses) based on probabilistic sensitivity analysis. Abbreviations: DALYs–disability-adjusted life years; YLDs–years lived with disabilities; YLLs–years of life lost

[*] input parameters were based on national statistics and published literature

[¶] Input parameters were based on published literature and anecdotal evidence

[+] Input parameters were based on anecdotal evidence.

Among 117,575 snakebite victims who were indicated for antivenom treatment (95%CrI 73,790–175,390), there were 954 amputations (95%CrI 383–1,797) and 15,909 deaths (95%CrI 7,592–33,949) following snakebite envenoming. Mortality of snakebite envenoming was estimated at 2.49 per 100,000 population (95%CrI 1.19–5.32), ranging from the lowest in Thailand (0.006 per 100,000 population) to the highest in Lao PDR (14.04 per 100,000 population) (**Fig 2** and **S2 Table**).

It was estimated that 80,813 snakebite victims in ASEAN (31% of victims who were indicated for antivenom treatment) were treated with antivenom, ranging from the lowest in Lao PDR (4.2%) to the highest in Thailand (99.9%) (**Fig 3**).

## Economic burden of snakebite in ASEAN

Annual economic burden of snakebite in ASEAN was estimated at 2.5 billion USD (95%CrI 1.2–5.4 billion USD) which was equivalent to 0.09% (95%CrI 0.04–0.20%) of the GDP (**Table 2** and **Fig 4**). The total costs of snakebite included direct medical costs of 69.0 million USD (95%CrI 49.0–94.8 million USD), direct non-medical costs of 6.5 million USD (95%CrI 4.2–10.3 million USD), and indirect costs of 2.4 billion USD (95%CrI 1.1–5.3 billion USD). The estimated economic burden of snakebite ranged from the lowest in Malaysia (2 million USD) to the highest in Indonesia (1.9 billion USD).

The total economic burden of 2.5 billion USD was broken down into hospitalization costs (59.7 million USD; 2.4% of the total economic burden), antivenom-related costs (9.2 million USD; 0.4%), amputation costs (0.1 million USD, 0.005%), transportation costs (3.1 million USD, 0.1%), food costs (3.4 million USD, 0.1%), productivity losses of victims and relatives during snakebite episode (16.1 million USD, 0.6%), and productivity losses due to premature death (2.4 billion USD, 96.4%).

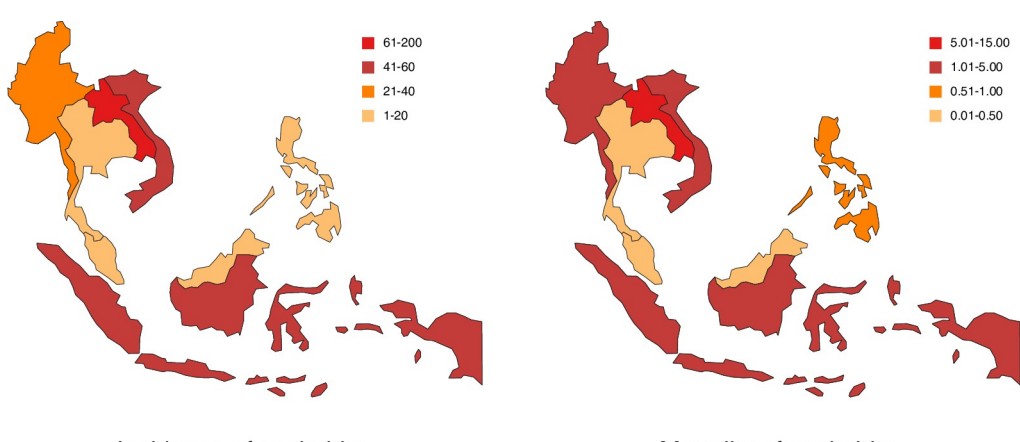

**Fig 2. Estimated annual epidemiological burden of snakebite in ASEAN countries.** The estimated incidence of snakebite ranged from the lowest in Malaysia (10.68 per 100,000 population) to the highest in Lao PDR (200.00 per 100,000 population). The estimated mortality of snakebite envenoming ranged from the lowest in Thailand (0.006 per 100,000 population) to the highest in Lao PDR (14.04 per 100,000 population). Main sources of information were national statistics and published research for the burden estimation of Malaysia, Thailand, and Myanmar. Published research and anecdotal evidence (local data, and expert opinion) were the main sources of information for the burden estimation of Vietnam, and Lao PDR. Anecdotal evidence was the only source of information for the burden estimation of Indonesia, and Philippines. Made with Natural Earth. Free vector and raster map data @ naturalearthdata.com.

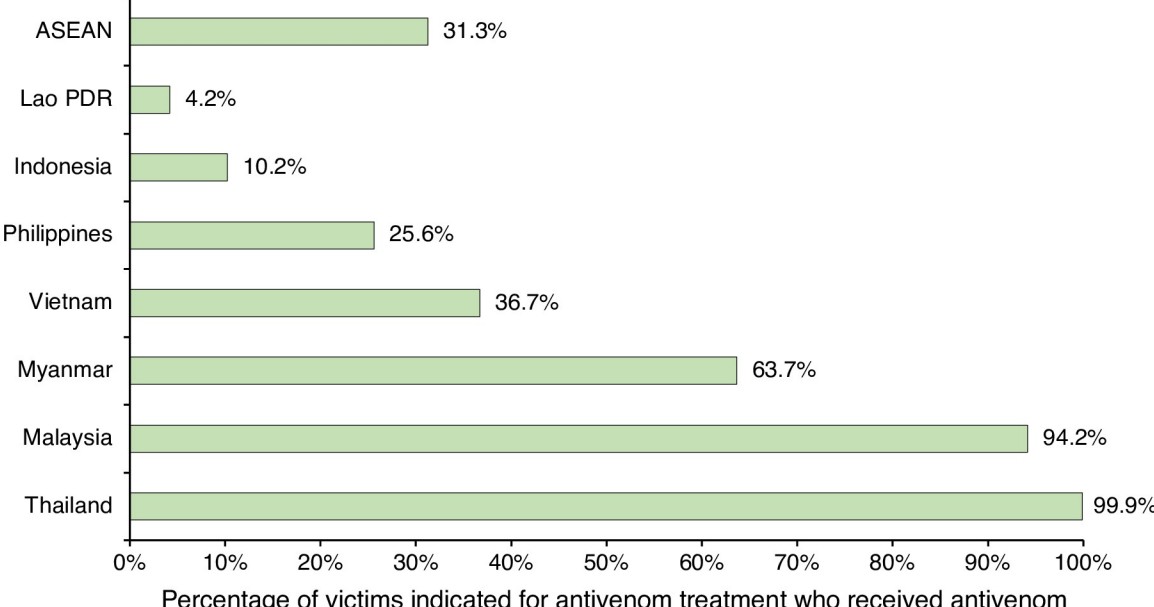

**Fig 3. Estimated proportions of snakebite victims treated with antivenom in ASEAN countries.** Percentages are estimated from number of snakebite victims treated with antivenom divided by total number of snakebite victims with systemic envenoming who need antivenom; Main sources of information were national statistics and published research for the burden estimation of Malaysia, Thailand, and Myanmar. Published research and anecdotal evidence (local data, and expert opinion) were the main sources of information for the burden estimation of Vietnam, and Lao PDR. Anecdotal evidence was the only source of information for the burden estimation of Indonesia, and Philippines.

**Table 2. Estimated annual economic burden (x1,000 USD) of snakebite in ASEAN countries.**

| | Direct medical costs, x1,000 USD | | | Direct non-medical costs, x1,000 USD | | Indirect costs, x1,000 USD | | Total costs, x1,000 USD | Total costs, % of GDP |
|---|---|---|---|---|---|---|---|---|---|
| | Healthcare costs | Antivenom-related costs | Amputation costs | Transportation costs | Additional food costs | Productivity losses during Snakebit episode | Productivity losses due to Premature death | | |
| Malaysia[*] | 754 (620–932) | 475 (249–758) | 0 | 38 (34–42) | 29 (23–40) | 366 (289–484) | 622 (0–1,866) | 2,284 (1,380–3,736) | 0.001% (0.000–0.001%) |
| Thailand[*] | 2,027 (1,615–2,531) | 1,176 (844–1,506) | 0.2 (0–0.6) | 58 (54–64) | 50 (37–67) | 925 (702–1,190) | 762 (381–1,333) | 4,999 (3,861–6,260) | 0.001% (0.001–0.001) |
| Indonesia[+] | 51,836 (36,900–70,844) | 4,129 (3,727–4,520) | 100 (44–178) | 1,579 (1,431–1,738) | 1,442 (1,027–1,970) | 8,752 (6,506–11,566) | 1,922,241 (914,489–4,110,887) | 1,988,891 (975,513–4,202,049) | 0.178% (0.087–0.375%) |
| Philippines[+] | 444 (338–578) | 147 (130–162) | 1 (1–2) | 63 (52–76) | 46 (35–60) | 638 (518–793) | 81,905 (40,762–163,735) | 83,244 (42,165–165,246) | 0.022% (0.011–0.044%) |
| Vietnam[⁋] | 3,208 (1,090–7,137) | 1,094 (447–1,210) | 0 | 853 (299–1,874) | 1,463 (494–3,264) | 3,801 (1,320–8,251) | 257,594 (76,180–690,928) | 268,013 (82,106–710,764) | 0.102% (0.031–0.271%) |
| Lao PDR[⁋] | 55 (42–71) | 27 (23–32) | 12 (2–34) | 13 (12–15) | 16 (13–20) | 427 (361–501) | 80,031 (40,573–159,767) | 80,583 (41,188–160,291) | 0.443% (0.227–0.882%) |
| Myanmar[*] | 1,382 (1,047–1,815) | 2,159 (1,910–2,425) | 0 | 474 (417–526) | 394 (303–516) | 1,208 (952–1,551) | 73,569 (44,703–131,172) | 79,186 (50,302–136,615) | 0.104% (0.066–0.180%) |
| Total | 59,706 (41,652–83,950) | 9,208 (7,329–10,613) | 114 (46–215) | 3,078 (2,299–4,335) | 3,441 (1,932–5,938) | 16,117 (10,648–24,335) | 2,416,724 (1,117,087–5,259,687) | 2,507,199 (1,196,516–5,384,962) | 0.091% (0.043–0.195%) |

Estimates are presented as base-case estimates (x 1000 USD) with their 95% credibility interval (in parentheses) based on probabilistic sensitivity analysis. Costs are presented as 2019 USD where 1 USD = 14,147.67 Indonesian Rupees = 51.80 Philippine Pesos = 23,050.24 Vietnamese Dong = 8,679.41 Lao Kip = 1,518.26 Myanmar Kyat. Abbreviation: GDP–gross domestic product; USD—US Dollar

[*] input parameters were based on national statistics and published literature

[⁋] Input parameters were based on published literature and anecdotal evidence

[+] Input parameters were based on anecdotal evidence.

## Disease burden of snakebite in ASEAN

We estimated an annual disease burden of snakebite in ASEAN of 391,979 DALYs (95%CrI 187,261–836,559), which was equivalent to 61 DALYs per 100,000 population (95%CrI 29–131) (**Fig 4** and **Tables 1** and **S2**). The estimated disease burden of snakebite involved 391,154 YLLs due to death from snakebite envenoming (95%CrI 186,491–835,263; 99.8% of the total DALYs), 330 YLDs for snakebite episode (95%CrI 154–613; 0.08%), and 495 YLDs for amputation (95%CrI 175–1,049; 0.13%). DALYs lost due to snakebite ranged from the lowest in Malaysia (52 DALYs) to the highest in Indonesia (262,888 DALYs).

## Comparison of economic and disease burden per victim with snakebite envenoming across countries

Economic and disease burden per victim with snakebite envenoming was compared across ASEAN countries (**S3 Table**). Mortality rate of snakebite envenoming ranged from the lowest in Thailand (0.001) to the highest in Lao PDR (0.332). Amputation rate of snakebite envenoming ranged from the lowest in Malaysia, Vietnam, and Myanmar (0.000) to the highest in Lao

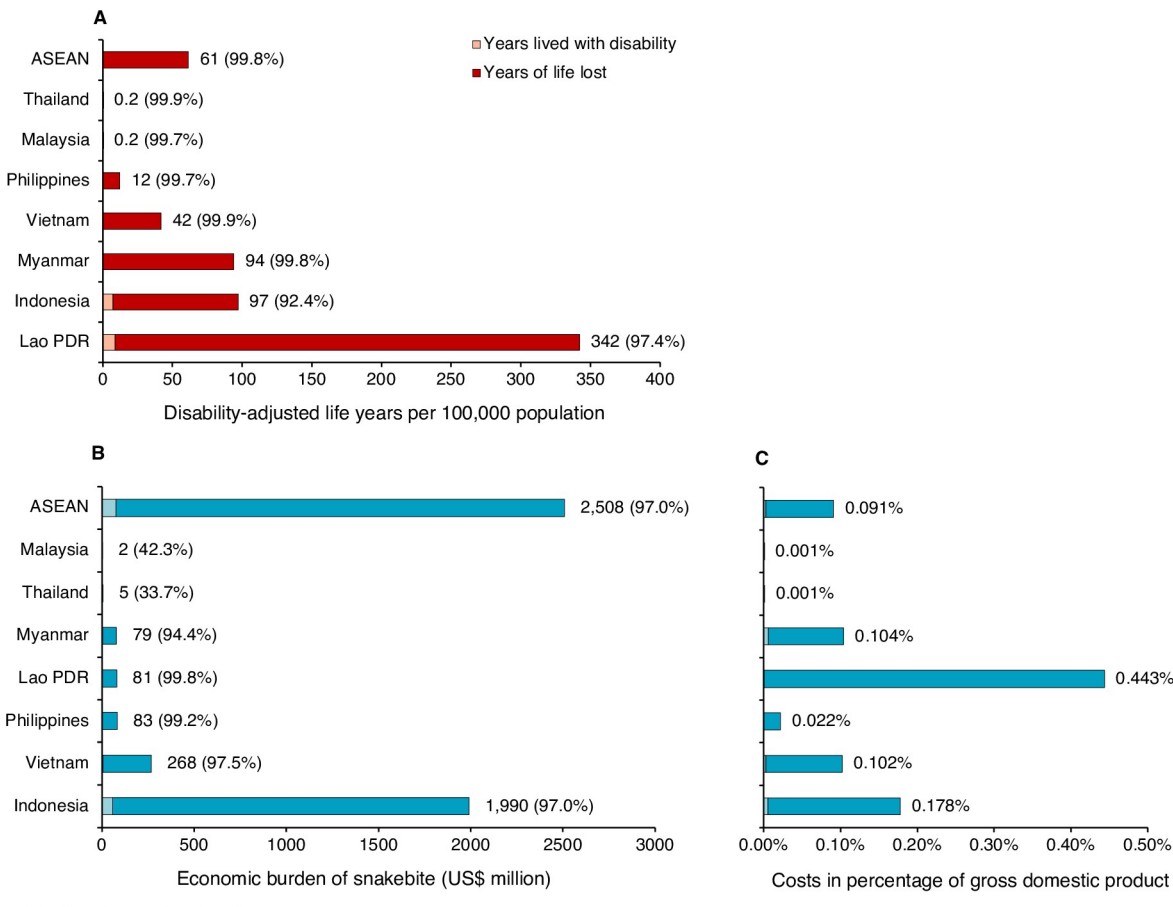

**Fig 4. Estimated annual economic and disease burden of snakebite in ASEAN countries.** (A) Disease burden of snakebite; data in parentheses are the percentages of disease burden attributable to years of life lost. (B) Costs in million USD; data in parentheses are the percentages of economic burden attributable to indirect costs. (C) Costs in percentage of gross domestic product; Main sources of information were national statistics and published research for the burden estimation of Malaysia, Thailand, and Myanmar. Published research and anecdotal evidence (local data, and expert opinion) were the main sources of information for the burden estimation of Vietnam, and Lao PDR. Anecdotal evidence was the only source of information for the burden estimation of Indonesia, and Philippines. Costs are presented as 2019 USD where 1 USD = 14,147.67 Indonesian Rupees = 51.80 Philippine Pesos = 23,050.24 Vietnamese Dong = 8,679.41 Lao Kip = 1,518.26 Myanmar Kyat. Abbreviation: GDP–gross domestic product; USD—US Dollar.

PDR (0.047). DALYs lost due to snakebite envenoming per victim ranged from the lowest in Thailand (0.02 DALYs per victim) to the highest in Lao PDR (8.10 DALYs per victim). Total costs of snakebite envenoming per victim ranged from the lowest in Thailand (861 USD per victim) to the highest in Philippines (47,072 USD per victim).

## Sensitivity analysis

One-way sensitivity analysis found that influential parameters for economic and disease burden were discount rate, probability of death due to snakebite envenoming, relative risk of death when antivenoms are not available, probability of systemic envenoming indicated for antivenom treatment, incidence of snakebite, and length of stay of victims indicated for antivenom treatment (**S1** and **S2** Figs). When PTSD was incorporated in the model in scenario analysis, the model estimated that there would be 10,293 cases of PTSD (95%CrI 4,651–20,954) with disease burden of 17,458 YLDs (95%CrI 5,869–40,035 YLDs) and productivity

losses of 12.7 million USD (95%CrI 4.7–27.9 million USD) (**S4 Table**). PTSD following snake-bite was found to slightly increased the economic (total costs of 2.52 billion USD; 0.5% increase) and disease burden (405,102 DALYs; 4.5% increase).

## Discussion

To achieve the goal set by the WHO to halve burden of snakebite by 2030, countries should know their current economic and disease burden of snakebite to understand their current standpoint. To our understanding, this is the first study conducted to estimate the economic and disease burden of snakebite in Southeast Asia. The annual economic and disease burden of snakebite in seven ASEAN countries were estimated using a decision analytic model incorporating input parameters from various sources including published literature and local sources to estimate the burden of all snakebite victims regardless of their treatment seeking behavior.

We estimated that annually there were 242,648 snakebite victims (95%CrI 209,810–291,023) of which 15,909 victims (95%CrI 7,592–33,949) were dead and 954 victims (95%CrI 383–1,797) were amputated. The estimated number of snakebite victims and deaths were comparable to the previous estimates in 2007 of approximately 234,000–1,410,000 snakebite victims and 700–18,000 deaths [1]. Annual disease burden of snakebite was estimated at 391,979 DALYs (95%CrI 187,261–836,559). Total costs of snakebite were estimated at 2.5 billion USD (95%CrI 1.2–5.4 billion USD) which were equivalent to 0.09% (95%CrI 0.04–0.20%) of the region's GDP. The share of the estimated economic burden from snakebite of the country's GDP ranged from 0.001% in Malaysia to 0.443% in Lao PDR which were remarkably high compared to less than 0.001%. in Iran and Burkina Faso and 0.016% in Sri Lanka [6–9]. The estimated disease burden of snakebite of 391,979 DALYs in seven ASEAN countries (61 DALYs per 100,000 population) was low compared to the previous estimates of 319,874 DALYs in 16 Western African countries (approximately 93 DALYs per 100,000 population) [11] and 1,029,209 DALYs in 41 Sub-Saharan African countries (approximately 120 DALYs per 100,000 population) [10]. This could be partly explained by the differences in the incidence and mortality of snakebite and access to antivenom treatment. Compared to the disease burden of neglected tropical diseases in seven ASEAN countries that were estimated in the Global Burden of Disease 2019 study, snakebite was the second highest burden ranking below dengue (909,899 DALYs) (**S3 Fig**). The disease burdens of malaria (72,844 DALYs) and rabies (66,525 DALYs) were much lower than snakebite [68].

In Malaysia and Thailand, >90% of victims indicated for antivenom could access to it. In contrast, remarkably lower proportions were demonstrated in Lao PDR, Indonesia, Philippines, Vietnam, and Myanmar of which 4–64% antivenom indicated victims were treated with antivenoms. These victims either sought traditional healers or were treated in healthcare facilities but did not receive antivenom due to inadequate supply of antivenom. Consequently, most deaths from snakebite envenoming (99.9%) in ASEAN were from Indonesia, Philippines, Vietnam, Lao PDR, and Myanmar which contributed to high economic and disease burden of premature death from snakebite envenoming. We found that more than 95% of the estimated economic and disease burden was attributed to premature deaths. Treating all snakebite victims who need antivenoms in these countries would save their lives which would result in a tremendous decrease in the burden of snakebite in ASEAN. However, increasing access to antivenom was not only about producing antivenoms but the whole surrounding supporting and management system especially the information system to inform decision making and logistics to efficiently deliver antivenoms even to the farthest healthcare facilities. We previously assessed the situation of snakebite in ASEAN and provided the potential opportunities

to improve situation of snakebite in ASEAN to meet the WHO's target of halving snakebite mortality and morbidity by 2030. These potential opportunities included accurate estimation of antivenom demand, rigorous regulations of antivenom, strengthening the supply chain system, raising public awareness about the importance of treating snakebite envenoming by healthcare professionals, strengthening the health system to ensure appropriate snakebite management and rational use of antivenoms, and expanding collaboration of local and international stakeholders and funders [5].

There were few important limitations of this study worth mentioning. Firstly, Cambodia was not included in this study because we were not able to identify published literature and key informants that could be utilized to estimate the burden of snakebite in Cambodia. It is important to note that Cambodia is one of the countries that imported antivenoms from Thailand, indicating that there were snakebite victims in this country [5]. Secondly, consequences of snakebite included in the model and its sensitivity analysis were limited to death, amputation, and PTSD. Other disabilities such as blindness, malignant ulcers, and pregnancy loss were not included due to a lack of empirical evidence in ASEAN [13]. This warrants future studies in ASEAN to evaluate all relevant consequences and disabilities and associated costs of snakebite to allow better estimation of burden of snakebite. Lastly, there was no nation-wide community and hospital study to comprehensively collect the number of snakebite victims in some of the included countries. Hence, input parameters must be estimated based on non-national studies, local data, and expert opinions, resulting in a wide range of the estimated economic and disease burden of snakebite in ASEAN. This is especially relevant in Lao PDR and Indonesia where snakebite incidences were very high and estimated by local experts. Nevertheless, our findings suggested that there was high burden of snakebite despite the availability of domestically produced antivenoms in the region. We emphasized the importance of funding research to perform a comprehensive data collection on epidemiological and economic burden of snakebite to eventually reveal the true burden of snakebite in ASEAN. These data will yield more accurate information on burden of snakebite to guide decision making in not only the ASEAN but also the WHO to develop global strategies to tackle the problem of snakebite.

## Conclusion

Annual production of 290,000 vials of antivenom in ASEAN were given to only 31% of victims who were indicated for antivenom treatment. Our estimates highlighted the high economic and disease burden of snakebite in ASEAN despite the availability of domestically produced antivenoms. Almost all of the estimated economic and disease burdens were attributed to premature deaths from snakebite envenoming which suggested that the remarkably high burden of snakebite could be averted, especially in countries where large proportions of victims who needed antivenom were not treated with geographically appropriate antivenoms. Strategies should be developed with the goal to improve health outcomes of snakebite victims. However, strategies used to achieve this goal are likely to be complex and different across countries depending on each country's context and situation such as accurate informatics, rigorous regulations of antivenoms, efficient supply chain, rational use of antivenoms, appropriate treatment seeking behaviors, and good governance to support a strong healthcare system.

## Supporting information

**S1 Appendix. Justification of input parameters.**
(DOCX)

**S2 Appendix. Estimation of economic and disease burden of post-traumatic stress disorder following snakebite envenoming.**
(DOCX)

**S1 Table Input parameters for estimating economic and disease burden of snakebite in ASEAN countries.**
(DOCX)

**S2 Table. Estimated annual epidemiological and disease burden of snakebite in 2019 in ASEAN countries.**
(DOCX)

**S3 Table. Estimated annual epidemiological and disease burden of snakebite envenoming per case in ASEAN countries.**
(DOCX)

**S4 Table. Estimated annual economic and disease burden of post-traumatic stress disorder following snakebite.**
(DOCX)

**S1 Fig. One-way sensitivity analysis of economic burden.**
(DOCX)

**S2 Fig. One-way sensitivity analysis of disability-adjusted life years (DALYs) of snakebite.**
(DOCX)

**S3 Fig. Comparison of annual disease burden of neglected tropical diseases in ASEAN.**
Estimated disease burden of snakebite from this study (shown in purple) was compared to the disease burden of neglected tropical diseases in seven ASEAN countries that were estimated in the Global Burden of Disease 2019 study.
(DOCX)

## Acknowledgments

The authors would like to thank members of the Pan ASEAN Antivenom (PAAV) consortium for their information and insightful recommendations on this work. The authors would like to thank Luke Joseph Schwerer for his diligent proofreading of this article.

## Author Contributions

**Conceptualization:** Chanthawat Patikorn, Suthira Taychakhoonavudh, Nathorn Chaiyakunapruk.

**Data curation:** Chanthawat Patikorn.

**Formal analysis:** Chanthawat Patikorn, Suthira Taychakhoonavudh, Nathorn Chaiyakunapruk.

**Funding acquisition:** Ahmad Khaldun Ismail, Iekhsan Othman, Suthira Taychakhoonavudh, Nathorn Chaiyakunapruk.

**Investigation:** Chanthawat Patikorn, Iekhsan Othman, Suthira Taychakhoonavudh, Nathorn Chaiyakunapruk.

**Methodology:** Chanthawat Patikorn, Suthira Taychakhoonavudh, Nathorn Chaiyakunapruk.

**Project administration:** Chanthawat Patikorn, Syafiq Asnawi Zainal Abidin, Iekhsan Othman, Suthira Taychakhoonavudh, Nathorn Chaiyakunapruk.

**Resources:** Jörg Blessmann, Myat Thet Nwe, Patrick Joseph G. Tiglao, Taksa Vasaruchapong, Tri Maharani, Uyen Vy Doan, Ahmad Khaldun Ismail.

**Supervision:** Iekhsan Othman, Suthira Taychakhoonavudh, Nathorn Chaiyakunapruk.

**Validation:** Jörg Blessmann, Myat Thet Nwe, Patrick Joseph G. Tiglao, Taksa Vasaruchapong, Tri Maharani, Uyen Vy Doan, Ahmad Khaldun Ismail.

**Visualization:** Chanthawat Patikorn, Suthira Taychakhoonavudh, Nathorn Chaiyakunapruk.

**Writing – original draft:** Chanthawat Patikorn, Suthira Taychakhoonavudh, Nathorn Chaiyakunapruk.

**Writing – review & editing:** Chanthawat Patikorn, Jörg Blessmann, Myat Thet Nwe, Patrick Joseph G. Tiglao, Taksa Vasaruchapong, Tri Maharani, Uyen Vy Doan, Syafiq Asnawi Zainal Abidin, Ahmad Khaldun Ismail, Iekhsan Othman, Suthira Taychakhoonavudh, Nathorn Chaiyakunapruk.

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
