## [Decision Letter · Decision Letter 0]

27 Jun 2022

Dear Nathorn 

Thank you very much for submitting your manuscript "Estimating economic and disease burden of snakebite in ASEAN countries using a decision analytic model" for consideration at PLOS Neglected Tropical Diseases. As with all papers reviewed by the journal, your manuscript was reviewed by members of the editorial board and by several independent reviewers. The reviewers appreciated the attention to an important topic. Based on the reviews, we are likely to accept this manuscript for publication, providing that you modify the manuscript according to the review recommendations.

Sincerely,

Indika Gawarammana

Deputy Editor

Indika Gawarammana

Deputy Editor

Please attend to the comments made by the reviewers .

Reviewer's Responses to Questions

**Key Review Criteria Required for Acceptance?**

**Methods**

-Are the objectives of the study clearly articulated with a clear testable hypothesis stated?

-Is the study design appropriate to address the stated objectives?

-Is the population clearly described and appropriate for the hypothesis being tested?

-Is the sample size sufficient to ensure adequate power to address the hypothesis being tested?

-Were correct statistical analysis used to support conclusions?

-Are there concerns about ethical or regulatory requirements being met?

Reviewer #1: I am not a Health Economist and my review is therefore limited in its scope. I sincerely hope that the Editor has sought a review from a suitable qualified Health Economist

-The objectives of the study were clearly articulated and a clear testable hypothesis was stated.

-The study design is appropriate to address the stated objectives - but does necessarily (due to lack of primary data) make many assumptions. These assumptions include data on:

 - incidence of PTSD following snake envenoming - the authors need to add a reference for this because I don't know any 

 reliable study on this topic

 - days lost post-envenoming - how was this calculated? This needs to be added to the supplementary data

-The populations examined in this study are clearly described and appropriate.

-The sample size is sufficient to ensure adequate power.

-Appropriate statistical analysis was used to support conclusions.

-Based upon the manuscript I believe that there no concerns about ethical or regulatory requirements

Reviewer #2: The objective is clearly stated. The study design is appropriate.

Only one-year incidence of snakebite was used for the calculation. When available, average incidence from 3-5 years should be used.

Reviewer #3: Estimating economic and disease burden of snakebite in ASEAN countries using a

decision analytic model 

It has been very stimulating and enlightening to review this very interesting and timely paper on economic and disease burden of snakebite in ASEAN countries. The study has clear objectives which have been achieved with a sound methodological design. 

I have a few minor comments:

1. The authors need to explain the rationale behind the choice of the countries selected for this analysis. While it may have been done based on previous literature, were any recent data considered to justify the selection? In addition, excluding Cambodia needs a stronger justification.

2. The decision analytic model used in this study, assumes that complications leading to disability occur only in snakebites that require anti-venom treatment. However, there is evidence to support that both physical and psychological complications can arise in any snakebite experience independent of envenomation. The authors need to consider this phenomenon and improve the model suitably.

3. Although disability due to snakebite was confined to amputations in this study, there is evidence that many other physical complications and residual health problems result from snakebite. This phenomenon and the difficulties associated with considering them for this analysis need to be discussed.

**Results**

-Does the analysis presented match the analysis plan?

-Are the results clearly and completely presented?

-Are the figures (Tables, Images) of sufficient quality for clarity?

Reviewer #1: The results are clearly presented, both in the manuscript and in the supplemental material

Reviewer #2: Due to the uncertain incidence in some countries, it is interesting to compare the data that are independent of incidence among different countries, e.g., death rate, amputation rate, average medical costs by case and average costs from productivity loss by case. This information is helpful to formulate an appropriate policy for each country.

Reviewer #3: The analyses are appropriate and the results are presented comprehensively.

**Conclusions**

-Are the conclusions supported by the data presented?

-Are the limitations of analysis clearly described?

-Do the authors discuss how these data can be helpful to advance our understanding of the topic under study?

-Is public health relevance addressed?

Reviewer #1: (No Response)

Reviewer #2: Line 295-296: The ASEAN burden is lower than that of Africa. Why the authors discuss that it is noticeably high?

Reviewer #3: The conclusions are appropriate.

However, a stronger conclusion towards prevention, improving access to anti-venom and better management of snakebite will help to inform policy in the ASEAN region.

**Editorial and Data Presentation Modifications?**

Reviewer #1: This important manuscript address a greatly under-researched and greatly-needed domain in the field of snakebite and I congratulate the authors for their efforts in acquiring data from very diverse sources.

The manuscript is well structured but would benefit from extensive editing on English phrasing and spelling.

Line 106 - there are so few papers examining the HE costs of snakebite that I was surprised to see that the PLOS paper on the HE burden of snakebite in Burkina Faso was not included - it should be (Ahmed S, Koudou GB, Bagot M, Drabo F, Bougma WR, Pulford C, Bockarie M, Harrison RA. Health and economic burden estimates of snakebite management upon health facilities in three regions of southern Burkina Faso. PLoS Negl Trop Dis. 2021 Jun 21;15(6)). The absence of this paper suggests that the authors need to run an updated extensive review of the global literature for snakebite HE studies.

If the authors are sufficiently confident in their HE data and analysis, I think the title should be revised to increase its impact and likelihood of being read by Public Health decision makers - perhaps something like 'Health economic analysis estimates that the 391,979 DALY annual disease burden of snake envenoming in the ASEAN region costs USD 2.5 billion'. 

I think the manuscript would benefit from an expansion of the overly brief discussion. This could include implications to WHO and each ASEAN country to meet the target of halving snakebite mortality and morbidity by 2030 (more primary research delivering data with fewer assumptions; cost benefits of managing snakebite better; funding implications - by Governments and WHO to meet this target; etc)

The conclusion would benefit by a very clear statement that the annual delivery of 290,000 vials of antivenom (apprx 42,000 treatments) is less than half that needed to treat the 117,575 victims - and describe the medical, societal, USD and GDP cost benefits of meeting the cost of delivery this expanded volume of antivenom. 

The authors could also make the important point, perhaps after the 'limitations' section) that funding of primary research would deliver much needed snakebite HE data that would reduce the data assumptions (that had to be made to complete this analysis) and yield accurate HE data for ASEAN countries, and WHO, to guide their decision making.

Reviewer #2: (No Response)

Reviewer #3: Minor Revision

**Summary and General Comments**

Reviewer #1: This is an important paper that adds valuable new information on the disease and health economic burden caused by snakebite. It could, but shouldn't, be criticised for the many data assumptions that were needed to populate their analytical tree - because the data was the best available and the results and public health implications therefrom are very important.

Reviewer #2: The knowledge of the disease burden in this area will be very helpful for the policy makers.

1. The incidence and burden are markedly heterogeneous among different countries suggesting that the policy changes should be country-specific.

2. The burden is largely contributed by the countries where the snakebite incidences were very high and estimated by local experts. This limitation should be addressed.

Reviewer #3: Overall this is a very good attempt at quantifying the burden of snakebite in the ASEAN region and will add significantly to the literature on snakebite.

PLOS authors have the option to publish the peer review history of their article (what does this mean?). If published, this will include your full peer review and any attached files.

Reviewer #1: No

Reviewer #2: No

Reviewer #3: No

Figure Files:

Data Requirements:

Reproducibility:

References

---

## [Decision Letter · Decision Letter 1]

30 Aug 2022

Dear Nathorn

We are pleased to inform you that your manuscript 'Estimating economic and disease burden of snakebite in ASEAN countries using a decision analytic model' has been provisionally accepted for publication in PLOS Neglected Tropical Diseases.

Best regards,

Indika Gawarammana

Section Editor

Reviewer's Responses to Questions

**Key Review Criteria Required for Acceptance?**

**Methods**

-Are the objectives of the study clearly articulated with a clear testable hypothesis stated?

-Is the study design appropriate to address the stated objectives?

-Is the population clearly described and appropriate for the hypothesis being tested?

-Is the sample size sufficient to ensure adequate power to address the hypothesis being tested?

-Were correct statistical analysis used to support conclusions?

-Are there concerns about ethical or regulatory requirements being met?

Reviewer #1: The authors have very satisfactorily addressed all my comments/suggestions

Reviewer #2: yes

**Results**

-Does the analysis presented match the analysis plan?

-Are the results clearly and completely presented?

-Are the figures (Tables, Images) of sufficient quality for clarity?

Reviewer #1: The authors have very satisfactorily addressed all my comments/suggestions

Reviewer #2: yes

**Conclusions**

-Are the conclusions supported by the data presented?

-Are the limitations of analysis clearly described?

-Do the authors discuss how these data can be helpful to advance our understanding of the topic under study?

-Is public health relevance addressed?

Reviewer #1: The authors have very satisfactorily addressed all my comments/suggestions

Reviewer #2: yes

**Editorial and Data Presentation Modifications?**

Reviewer #1: The authors have very satisfactorily addressed all my comments/suggestions

Reviewer #2: no

**Summary and General Comments**

Reviewer #1: The authors have very satisfactorily addressed all my comments/suggestions

Reviewer #2: I am satisfied with the revision

PLOS authors have the option to publish the peer review history of their article (what does this mean?). If published, this will include your full peer review and any attached files.

Reviewer #1: No

Reviewer #2: No

---

## [Editor Report · Acceptance letter]

13 Sep 2022

Dear Mr. Chaiyakunapruk,

We are delighted to inform you that your manuscript, "Estimating economic and disease burden of snakebite in ASEAN countries using a decision analytic model," has been formally accepted for publication in PLOS Neglected Tropical Diseases.

Best regards,

Shaden Kamhawi

co-Editor-in-Chief

Paul Brindley

co-Editor-in-Chief
